# New sampling strategy mitigates a solar-geometry-induced bias in sub-km vapour scaling statistics derived from imaging spectroscopy

Mark T. Richardson[1,2], David R. Thompson[1], Marcin J. Kurowski[1], Matthew D. Lebsock[1]

[1]Jet Propulsion Laboratory, California Institute of Technology, Pasadena, CA 91109, USA
[2]Department of Atmospheric Science, Colorado State University, Fort Collins, CO 90095, USA

*Correspondence to*: Mark Richardson (markr@jpl.nasa.gov)

**Abstract.** Upcoming spaceborne imaging spectrometers will retrieve clear-sky total column water vapour (TCWV) over land at horizontal resolution of 30—80 m. Here we show how to obtain, from these retrievals, exponents describing the power-law scaling of sub-km horizontal variability in clear-sky bulk planetary boundary layer (PBL) water vapour ($q$) accounting for realistic non-vertical sunlight paths. We trace direct solar beam paths through large eddy simulations (LES) of shallow convective PBLs, and show that retrieved 2-D water vapour fields are "smeared" in the direction of the solar azimuth. This changes the horizontal spatial scaling of the field primarily in that direction, and we address this by calculating exponents perpendicular to the solar azimuth, that is to say flying "across" the sunlight path rather than "towards" or "away" from the Sun. Across 23 LES snapshots, at SZA=60° the mean bias in calculated exponent is 38±12 % (95 % range) along the solar azimuth, while following our strategy it is 3±9 % and no longer significant. Both bias and root-mean-square error decrease with lower SZA. We include retrieval errors from several sources including: (1) the Earth Surface Mineral Dust Source Investigation (EMIT) instrument noise model, (2) requisite assumptions about the atmospheric thermodynamic profile, and (3) spatially nonuniform aerosol distributions. By only considering the direct beam we neglect 3-D radiative effects, such as light scattered into the field of view by nearby clouds. However, our proposed technique is necessary to counteract the direct-path effect of solar geometries and obtain unique information about sub-km PBL $q$ scaling from upcoming spaceborne spectrometer missions.

## 1 Introduction

Spatial scaling in the variability of atmospheric properties such as water vapour ($q$) can be characterised via structure functions, with the $n^{\text{th}}$ order structure function of a field $f(x)$, $S_n$ defined as:

$$S_n(r) = E[(f(x) - f(x + r))^n] \tag{1}$$

Where $E[]$ is the expected value, $x$ a location and $r$ a separation between points. Fields of temperature ($T$), $q$ and wind-speed are commonly well-modelled by a power law:

$$S_n(r) \propto r^{\zeta_n} \tag{2}$$

Such that $\zeta_n$ is the log-log gradient of $S_n$ as a function of $r$. There is strong motivation to quantify and understand these exponents and the ranges $\Delta r$ within which they are valid, and here we specify second-order structure functions $S_2$ with exponent $\zeta_2$, describing variance scaling. This is related to the commonly-referenced Fourier power spectrum exponent $\beta$:

$$\beta = -(\zeta_2 + 1) \tag{3}$$

One motivation for obtaining these exponents is that climate model sub-grid variability in $q$ is strongly linked to cloud formation (Golaz et al., 2002; Perraud et al., 2011; Sommeria and Deardorff, 1977). In principle sub-grid variance can be tuned for each model setup, but scale-aware variance relationships allow a smooth and consistent transition between low (order ~hundreds of km) and high (order ~km) resolution models (Arakawa et al., 2011; Schemann et al., 2013).

At scales larger than model grid cells, observational estimates of variance scaling can also be used to assess model performance, as has been done using estimates of temperature ($T$) and $q$ from Atmospheric Infrared Sounder (AIRS) and airborne campaign data (Kahn et al., 2011).

Furthermore, the scaling exponents are related to the physical processes that generate the cascade of turbulent eddies in the atmosphere. For example, while mean-scale statistics retrieved by AIRS are approximately isotropic in the horizontal, there

are differences in scaling between the horizontal and vertical (Pressel and Collins, 2012). Scaling following $\zeta_2=2/3$ is predicted for a passive tracer in the inertial range of three-dimensional locally isotropic turbulence following Kolmogorov theory. Accounting for the buoyancy effects can strongly modify that scaling (Bolgiano, 1959; Obukhov, 1959), with different exponents expected for the velocity ($\zeta_2=6/5$-$7/5$) and scalars ($\zeta_2=2/5$), as shown in, e.g. Kunnen et al. (2008), Wroblewski et al. (2010) or Boffetta et al. (2012). Exponents of 7/5 have been commonly measured for vertical wind profiles from dropsondes

(Lovejoy et al., 2007), and values typically near $\zeta_2$ of 6/5 for horizontal water vapour in non-convective areas, and $\zeta_2=0.72$ in convective areas have been determined from airborne lidar retrievals over horizontal ranges of up to 100 km (Fischer et al., 2012, 2013), meanwhile the sub-km regime remains undermeasured.

Of particular interest for modellers are the existence of "scale breaks", distances at which the exponents change such that a smooth transition between model resolutions may not be possible. These have been calculated to occur at distances over a

broad range of 10—1000 km (Bacmeister et al., 1996; Gage and Nastrom, 1985; Kahn et al., 2011; Pinel et al., 2012) and the range of scales has been speculated to be related to the size of convective systems (Dorrestijn et al., 2018) or changes in the nature of turbulence (Kurowski et al., 2015; Skamarock et al., 2014).

This study focusses on the estimation of horizontal scaling of clear-sky TCWV at sub-km scales, and specifically on the integrated PBL water vapour which we refer to as the partial column water vapour ($PCWV_{PBL}$). Recent work using airborne

data (Thompson et al., 2021) and large eddy simulation (LES) output (Richardson et al., 2021) has provided evidence that sub-km horizontal variability in total column water vapour (TCWV) is almost perfectly correlated with $PCWV_{PBL}$ variability, such that high-spatial-resolution retrievals of TCWV from visible and shortwave infrared (VSWIR) imaging spectrometers can provide unique information about PBL $q$ variability. This is not a statement that all water vapour is inside the PBL, rather that

on sub-km spatial scales, the variability in low-altitude water vapour dominates the horizontal column variability. It remains a challenge to disentangle variability from different heights within the PBL such as the sub-cloud layer or a conditionally unstable cloud layer, which may have different scaling properties.

In particular, the PBL depth is typically 1—2.5 km in these simulations, while we obtain variability statistics over horizontal ranges of under 1 km. The vertical averaging over a scale larger than the horizontal calculation means that the interpretation of the physical meaning of the derived exponents is challenging and may not be directly related to the theoretically derived exponents discussed above. This study aims only to determine whether the measurement problem of the solar path can be overcome, and leaves the physical interpretation out of scope. This point will be revisited in Section 4.

Several modern and upcoming missions obtain or will obtain VSWIR spectra that could allow retrieval of TCWV at horizontal resolutions from 20—100 m. Current examples include the Multi-Spectral Imager (MSI) on Sentinel-2 (Drusch et al., 2012), the PRecursore IperSpettrale della Missione Applicativa (PRISMA, Candela et al. (2016)) and DLR Earth Sensing Imaging Spectrometer (DESIS, Krutz et al. (2019)). Upcoming missions such as NASA's Earth Surface Mineral Dust Source Investigation (EMIT; Green and Thompson (2020)) and ESA's Copernicus Hyperspectral Imaging Mission for the Environment (CHIME; e.g. Rast et al. (2019)) will obtain horizontal resolution of order 30—80 m, and this study will assess performance assuming footprint sizes of 40—50 m, i.e. at the mid-point of that range.

This footprint selection was made based on the resolution of available LES output, and the analysis method is based on the output of retrievals developed specifically for EMIT, which primarily retrieves TCWV to allow atmospheric correction for its surface reflectance target observable. Other instruments such as PRISMA or MSI may have different error characteristics, but we treat retrieval errors in a general manner that could be expanded to these other instruments.

This study attempts to determine whether EMIT will be able to obtain $\zeta_2$ over 0.5—1 km after accounting for (1) random retrieval error, (2) systematic biases in retrieval mean and sensitivity and (3) solar zenith angle.

The PBL is of particular interest since it is the location where reflective low clouds form and in Dorrestijn et al. (2018)'s analysis, $\zeta_2$ varied more in the 850 hPa layer than the 300 hPa or 500 hPa layers. However, previous analyses have generally been restricted to far larger spatial ranges with Dorrestijn et al. (2018) referring to 55—165 km scale variance as occurring at the "tiny scale". Other examples at higher resolution generally use airborne measurements, with few calculations for separations under 1 km using onboard sensors (Cho et al., 1999), using lidar at 5—100 km (Fischer et al., 2013) or evaluating simulations with lidar for separations >11 km (Selz et al., 2017).

This study uses LES outputs and does not make any statements about the realism or cause of the LES output $\zeta_2$ values, but instead aims solely to identify and quantify retrieval biases and errors. Its greatest contribution is to demonstrate that directional calculation strategies can remove biases in estimates of $\zeta_2$ introduced by the direct-beam component of the solar path through the atmosphere.

We show that, in a set of 23 LES snapshots, the non-vertical direct-beam path prevents accurate retrieval of sub-km $\zeta_2$ when standard methods are naïvely applied. However, errors in $\zeta_2$ depend on the direction in which it is calculated relative to the

solar azimuth, and selecting the correct solar-aware direction eliminates the bias and therefore overcomes a fundamental barrier to VSWIR estimation of sub-km $q$ scaling. Diffuse sunlight is handled through a plane-parallel radiative transfer approximation, which means that complex 3-D radiative effects are neglected. In clear-sky areas near clouds, 3-D effects can brighten observed spectra (Várnai and Marshak, 2009), with induced biases of order ~0.25 % for VSWIR column $CO_2$

retrievals (Massie et al., 2021). The consequences for hyperspectral TCWV retrievals at 30—80 m horizontal resolution are not currently known, although the effect on retrieved $\zeta_2$ would depend on the spatial scaling of these TCWV biases. In Section 4 we propose a field experiment design to test our conclusions.Spaceborne spectrometers measure light that has passed along some path, $r_{\downarrow\uparrow}$, consisting of downward (Sun-to-surface, $r_\downarrow$) and upward (surface-to-sensor, $r_\uparrow$) components. Retrievals will respond to the path integrated water vapour (PIWV) between the surface and top of atmosphere (TOA):

$$PIWV = \int_{TOA}^{surface} q(r_\downarrow)\, dr + \int_{surface}^{TOA} q(r_\uparrow)\, dr \tag{4}$$

Meanwhile the commonly desired value is the vertically integrated water vapour, TCWV:

$$TCWV = \int_{surface}^{TOA} q(z)\, dz \tag{5}$$

For a horizontally uniform $q$ field (equivalent to plane-parallel in radiative transfer parlance) there is a simple geometric relationship between the two:

$$TCWV = PIWV_{uniform} / \left(\frac{1}{\mu} + \frac{1}{\mu_0}\right) \tag{6}$$

Where $\mu$ is the cosine of the sensor viewing zenith angle and $\mu_0$ the cosine of the solar zenith angle. Despite the fact that real $q$ fields vary horizontally such that Eq. (6) is not strictly true, the VSWIR community commonly assumes a plane-parallel atmosphere and reports the retrieved value as being TCWV (Carbajal Henken et al., 2015; Diedrich et al., 2015; Grossi et al., 2015; Nelson et al., 2016; Noël et al., 2004; Preusker et al., 2021). Here we calculate PIWV by tracing solar rays through 3-D

LES output but for consistency with VSWIR literature terminology, we relate it to an effective TCWV$_{eff}$:

$$TCWV_{eff} = PIWV / \left(\frac{1}{\mu} + \frac{1}{\mu_0}\right) \tag{7}$$

Our retrieved values, TCWV$_{ret}$, refer to estimates of this property. Figure 1 compares the true TCWV with TCWV$_{eff}$ derived from ray tracing through 3-D LES output when SZA=45° and solar azimuth is at 0°, meaning that the horizontal component of $r_\downarrow$ is in the negative $y$ direction. There is an apparent "smearing" from Figure 1(a) to Figure 1(b) in the $y$ direction, so we

refer to these solar-geometry induced changes in TCWV$_{eff}$ due to the horizontal variability in the $q$ field as the "solar smearing" effect (for a simplified illustration of the physical principles behind why our strategy is anticipated to reduce biases in $\zeta_2$, see Supplementary Figures 1—3).

This consequence of solar and view geometry is well known, for example Thompson et al. (2021) only used flight lines with very low SZA to minimise its effect in a study of $\zeta_2$. However, this severely limits viable data so here we suggest a new method

that exploits the directionality of the solar-smearing effect in order to perform such calculations across a wider range of conditions. This paper uses the solar-path-traced outputs of Richardson et al. (2021) to demonstrate that by calculating the structure function in a direction perpendicular to the solar azimuth, the bias in $\zeta_2$ is removed within the 23 LES snapshots

considered. To our knowledge, this is the first quantification of such a technique to mitigate solar-geometry-induced errors in spatial statistics of retrieved atmospheric properties.

If this result can be extended to the real-world, then upcoming high-spatial-resolution spaceborne VSWIR spectrometers will provide a breakthrough for analysis of moisture scaling across unprecedented spatial scales. Our retrievals are restricted to
clear sky areas over land, but this still represents a substantial advance on the current capacities of other instrument types. Lidar measurements avoid the solar-path issue, and can retrieve vertical information including above clouds, but current and anticipated spaceborne lidars do not offer VSWIR's fine spatial resolution or broad spatial coverage from a wide swath. Sounders such as infrared can profile the atmosphere, but have footprints that are too large for sub-km exploration.

Even airborne measurements, which offer far less coverage than spaceborne sensors, may suffer from their own challenges.
Evidence suggests that the tendency of flights to follow isobars rather than maintain altitude can introduce height variations that blend vertical variation into horizontal calculations and result in exponents similar to those predicted due to buoyancy's vertical effect (Lovejoy et al., 2004; Pinel et al., 2012).

The VSWIR sampling technique introduced here could greatly expand the range of conditions under which spatial scaling of water vapour can be quantified. In Section 2 we describe the LES output, simulated retrievals, and how retrieval errors and
solar path are accounted for in calculation of $\zeta_2$. Section 3 presents the results and Section 4 discusses and concludes.

## 2 Data and Methods

### 2.1 Large Eddy Simulation output

We use $q$ and cloud water ($q_c$) output from the five LES runs of shallow convection as in Richardson et al. (Richardson et al., 2021), with four cloudy cases (ARM, ARM_lsconv, BOMEX, RICO) and one case in which clouds do not form (DRY).
Simulations use two models, EULAG for the ARM cases (Prusa et al., 2008) and JPL-UCONN LES for the others (Matheou and Chung, 2014). Simulation setups are described in Richardson et al. (Richardson et al., 2021) and the associated references (Brown et al., 2002; Kurowski et al., 2020; Matheou and Chung, 2014; Siebesma et al., 2003; vanZanten et al., 2011), and while some simulations represent oceanic boundary layers we simply assume a land surface for the retrievals. Static reanalysis profiles from MERRA-2 (Gelaro et al., 2017) are appended above the LES domain, but the LES domains were shown to
capture horizontal variability in $q$ from analysis of LES output and airborne lidar profiles over the Pacific (Bedka et al., 2021). The 23 selected snapshots are labelled by their timestamp, e.g. DRY_7200s represents two hours into the DRY simulation. The LES output horizontal resolution ranges from $\Delta x$=20—50 m, here we degrade the $\Delta x$=20 m cases to $\Delta x$=40 m resolution to make the spatial difference statistics more consistent.

## 2.2 Generating retrieved TCWV fields

### 2.2.1 Emulator development

The methodology here applies the results of Richardson et al. (2021), which developed a TCWV retrieval emulator to rapidly generate $TCWV_{ret}$ fields given $TCWV_{eff}$ derived from 3-D LES $q$ fields. This emulator was developed from results of an observing system simulation experiment (OSSE). Our differential optical absorption spectroscopy (DOAS) retrieval requires an accurate representation of water vapour spectroscopy, so we selected the MODTRAN6.0 radiative transfer model for forward and inverse calculations (Berk et al., 2014, 2015). The retrievals were performed using the Imaging Spectrometer Optimal Fitting (ISOFIT) code (Thompson et al., 2018, 2019) with EMIT instrument characteristics and noise. ISOFIT simultaneously retrieves the surface reflectance spectrum ($\rho_s(\lambda)$), aerosol optical depth (AOD) and $TCWV_{ret}$ from radiance over $\lambda$=380—2500 nm. The retrieval includes a lookup table (LUT) that relates TCWV and AOD to radiance properties and this LUT is generated by radiative transfer using uniformly scaled $q(z)$ and aerosol extinction ($\beta_{ext}(z)$) profiles to match desired $TCWV_{eff}$ and AOD.

Some studies performed radiative transfer over complete LES fields, but we found that full-field simulations were too computationally expensive given our toolkit and requirements. For example, Gristey et al. (2019) performed 3-D radiative transfer simulations over full LES fields, but they were interested in broadband fluxes so could use lower spectral resolution (370 wavelengths versus >20,000 here). Furthermore, their LES output was smaller (average ~60k footprints versus ~500k here) and this work must consider numerous combinations of properties such as surface type and solar zenith angle.

To reduce computational expense, Richardson et al. (2021)'s OSSE selected 101 footprints from each LES snapshot and used their vertical profiles as MODTRAN6.0 input to generate the true forward radiance spectra. MODTRAN6.0 is a plane-parallel radiative transfer model which must assume a horizontally uniform atmosphere but accounts for solar and view geometry, so this step applies Eq. (6). From the 101 pairs of TCWV and $TCWV_{ret}$ for a given combination of surface type and SZA, a linear relationship between $TCWV_{eff}$ (in this case, $TCWV_{eff}$=TCWV) and $TCWV_{ret}$ was found:

$$TCWV_{ret} = a_1 TCWV_{eff} + a_2 + \epsilon \; , \tag{8}$$

Where $a_1$ and $a_2$ are the slope and intercept and $\epsilon$ a random sample from a Normal distribution whose standard deviation $\sigma_\epsilon$ quantifies the random retrieval error. The parameter $a_1$ represents the sensitivity $dTCWV_{ret}/dTCWV_{eff}$ and $a_2$ is related to the bias, although it is only equal to the bias if $a_1$=1. Fits at different timesteps from each LES simulation were not significantly different from each other, but parameters did differ between simulations. To account for this, emulators were generated separately for each LES simulation but footprints from all snapshots in each simulation were combined, resulting in a sample size of 303—707 to estimate $a_1$, $a_2$ and $\sigma_\epsilon$ in each case (parameter estimates stabilised around $N$=50).

Figure 2(a) shows the linear relationship from all ARM snapshots, while Figure 2(b—e) show how the parameters may vary with surface type, SZA, retrieval-assumed atmospheric profile shapes, and AOD.

Figure 2(b) shows that $a_2$ varies between a vegetation and a mineral surface. Surface type can vary greatly on sub-km scales and within-scene transitions between vegetation and mineral surfaces would introduce artificial variance in $TCWV_{ret}$

differences either side of transition boundaries, and potentially affect the derived $S_2$ and $\zeta_2$. Variations in $a_2$ are small when considering mixtures of vegetation or mixtures of mineral-urban surfaces, so we limit our structure-function analysis to mixed-vegetation or mineral-urban surfaces.

From Figure 2(c), $TCWV_{ret}$ is only weakly sensitive to SZA from 14° to 60°, the derived parameters do not differ significantly

and for SZA from 14—45° $TCWV_{ret}$ are extremely similar. From this we argue that we can use a single set of derived parameters for each LES case for SZA up to 60°.

Next, we note that absorption line broadening depends on thermodynamics, so argue that the structure of $q(z)$ can change $a_1$ and $a_2$. For example, changes in $q$ at a lower, warmer level will result in stronger changes in absorption at the edges of bands compared with changes in $q$ at a higher, cooler level. Figure 2(d) presents results from one test that support this argument,

when the atmosphere used in the retrieval's LUT is changed from midlatitude summer to tropical the parameters change. Similarly, substantial changes in the forward model $q(z)$ profiles changes the derived parameters (not shown). We note that Figure 2(a) contains results from different LES timesteps in which the atmospheric profiles evolved, but this evolution was too small to generate any detectable differences in the Eq. (8) parameters.

Finally, we considered the relationship between AOD and the parameters in Eq. (8). In all other panels AOD varied randomly

from 0.1—0.2 between footprints, but in Figure 2(e) the simulations are performed with AOD fixed across all footprints at either 0.05, 0.20 or 0.35. Changing AOD by 0.3 results in a change in $a_2$ equivalent to approximately 0.3 % of $TCWV_{eff}$. This is a smaller change than between common surface types (Figure 2(b)), and AOD varies far more smoothly at sub-km horizontal scales than surface type, so we anticipated that our results would be robust to some types of spatial aerosol variability. In Section 3.2 we show how vertical gradients of up to 0.3 AOD km$^{-1}$ can have a minor effect on derived exponents.

**2.2.2 Generating retrieved fields**

For SZA from 0—60° in increments of 15° the PIWV was calculated by ray tracing from the top of the atmosphere to centre of each surface grid cell, and then directly up with a solar azimuth of 0°. This geometry represents a nadir instrument view angle, and PIWV is then converted to $TCWV_{eff}$ via Eq. (7), which accounts for SZA. This $TCWV_{eff}$ is then used as input for Eq. (8), thereby generating $TCWV_{ret}$ fields for analysis.

As mentioned in Section 2.2.1, each separate LES case has a unique set of Eq. (8) parameters and the fit is generated from plane-parallel radiative transfer calculations. This means that horizontal variability in $q$ is not explicitly accounted for, but we argue that the derivation includes a range of $q(z)$ profiles, and there is no fundamental reason that $q(r_{\downarrow\uparrow})$ should introduce fundamental differences that should affect our Eq. (8) relationship. Therefore, we assume that the relationship between $TCWV_{ret}$ and PIWV will be captured by Eq. (4).


The same ray-traced calculation is repeated with cloud water $q_c$ to obtain cloud water path (CWP). Footprints are then flagged as cloudy or shaded when CWP>1×10$^{-3}$ mm for any of the selected SZAs so as to avoid changes in $\zeta_2$ due to changes in the footprints considered. This CWP is equivalent to approximately $\tau$>0.3 in a typical subadiabatic cloud (Szczodrak et al., 2001).

## 2.3 Calculation of spatial statistics and removal of random error

$S_2(r)$ is calculated from the LES field of $\text{TCWV}_{\text{ret}}$ in one horizontal direction at a time, by including all pairs of footprints separated by $r$ in that direction provided that neither of the footprints in the pair is flagged as cloudy or shadowed for any SZA. To calculate along the LES $x$ axis, each different $y$ location is effectively treated as a 1-D field and it's set of $f(x+r)-f(x)$ values are calculated, then all the 1-D subsets are concatenated and the reported $S_2(r)$ is the expectation value of this combined dataset. The horizontal directions are either along the $x$ axis ("perpendicular" to the sunlight) or along the $y$ axis (the "parallel" case). Figure 3 displays example $S_2(r)$ for clear-sky TCWV and their associated scaling parameters, i.e., $dln(S_2)/dln(r)$, showing that $\zeta_2$ will depend on the range of $r$ over which it is calculated. The variation of $\zeta_2$ with $\Delta r$ can be due to changes in physical processes in addition to imperfect process representation in the LES, such as nonphysical dissipation at separations smaller than several grid cells (Brown et al., 2002). This study is not concerned with the interpretation of $\zeta_2$ but rather with the accuracy with which it can be obtained, so we do not explore this further and follow Thompson et al. (2021) in calculating the fit over $r=0.5$—1 km.

We estimate and remove the measurement noise following the method of Richardson et al. (2021). They derived a structure-function-based method to estimate $\sigma_\varepsilon$ from the $\text{TCWV}_{\text{ret}}$ field, thereby allowing better estimation of $S_2$ by subtracting $2\sigma_\epsilon^2$ from Eq. (5). This method involves calculating $S_2$ in one horizontal direction at $r=\Delta x$ (i.e. separation of 1 grid cell, 40 m or 50 m), then smoothing the field by a factor of two in the perpendicular direction and recalculating $S_2$, which we label $S_{2,\times 2}$. At these small separations, $2\sigma_\epsilon^2 \gg a_1^2 S_2(r)$, such that $S_{2,\times 2} - S_2 \approx 2\sigma_\epsilon^2$.

We evaluate the sensitivity of our derived $\zeta_2$ to uncertainty in the retrieval emulator, when calculating $S_2$ from the retrieved fields the emulator changes the retrieved value $S_{2,ret}$ as:

$$S_{2,ret}(r) = a_1^2 S_2(r) + 2\sigma_\epsilon^2 \tag{9}$$

And the estimated $\zeta_{2,ret}$ is;

$$\zeta_2 = \frac{d \ln(a_1^2 S_2 + 2\sigma_\epsilon^2)}{d \ln(r)} \tag{10}$$

This shows that both biases in $a_1$ ("retrieval sensitivity") and the magnitude of random errors can change the derived $\zeta_2$. Richardson et al. (2021) showed that errors in $a_1$ were the largest source of uncertainty in the estimating the spatial standard deviation. We will therefore perform sensitivity tests for a range of $a_1$ values, and show only one $\sigma_\varepsilon$ case since results were insensitive to changes in $\sigma_\varepsilon$ up to a factor of four scaling.

The final question we address is whether the scaling exponent estimated at the very high spatial resolution of missions such as EMIT will be fundamentally different from that obtained by current sensors such as MERIS and MODIS that can provide TCWV at a nominal resolution near $\Delta x \approx 250$ m. Large-scale $\sigma_x$ narrows as spatial resolution coarsens, but it is not clear how this affects the spatial scaling properties considered here, so for this test we sequentially degrade a TCWV field from $\Delta x=50$ m to $\Delta x=250$ m and calculate $S_2$ and $\zeta_2$ at each resolution.

## 3 Results

### 3.1 Variation of horizontal $\zeta_2$ with height

Figure 4 shows how the $\zeta_2$ calculated for PCWV integrated up to different heights varies within the PBL, but that the value becomes fixed by the PBL top. That is, estimates made from TCWV refer to the value for $PCWV_{PBL}$, but this conceals vertical structure within the PBL. We can therefore confirm that our derived values are indeed representative of $\zeta_2$ derived from $PCWV_{PBL}$, but further work is needed to determine the precise utility of statistics of $PCWV_{PBL}$ and furthermore, we note that corresponding estimates of PBL height from other sources may be necessary to help interpret measurements of $PCWV_{PBL}$ scaling.

### 3.2 Retrieval errors and $\zeta_2$

The results from all sensitivity tests applied to the clear-sky TCWV in the ARM_18000s snapshot are shown in Figure 5, results are similar for other snapshots (not shown). Figure 3(a) shows that random errors that we estimate will be typical for EMIT result in substantial changes to calculated $\ln(S_2)/\ln(r)$, with a notable flattening over much of the $r$ range and an unacceptably large error in $\zeta_2$ derived from retrieved TCWV. Our error correction reduces the $\zeta_2$ error from 53.1 % to 1.4 %. Figure 5(b) shows that errors in the sensitivity $dTCWV_{ret}/dTCWV_{eff}$, which is the emulator parameter $a_1$ in Eq. (8), shift $S_2$ but

do not substantially affect the gradient over $r$=0.5—1 km, and this conclusion also applies in Figure 5(c) when biases in $a_1$ are combined with $\sigma_\varepsilon$ and the error correction is applied. These results show that even large biases in $a_1$, which contribute proportionally to estimates of errors in $\sigma_x$, do not have a substantial effect on derived scaling properties.

Figure 5(d) confirms that scaling properties are sensitive to the measurement resolution, with derived exponents varying unpredictably from 0.61—0.73 with resolution. For $\Delta x$=250 m there are only 5 points included in the regression. Furthermore,

more footprints will be partially cloudy, meaning that the samples included will be too small for robust estimation of $\zeta_2$. We did not apply a matched cloud mask for this test, and expect that this will lead to more uncertain estimates of $\zeta_2$. Tests across all 23 snapshots show that most return a higher, and more uncertain, $\zeta_2$ when spatial resolution is degraded (not shown). This points to new information being obtained from finer-spatial-resolution retrievals, provided that the slanted solar paths do not destroy the correspondence between true and retrieved $\zeta_2$.

The final panel Figure 5(e) shows a sensitivity test to strong spatial gradients in AOD. From Figure 2(e), a change of 0.3 in AOD results in a difference of 0.3 % in $TCWV_{ret}$ in ARM_18000s, and other tests found a smaller fractional response in DRY_7200s (not shown). We picked the ARM_18000s results as the worst case and allowed $TCWV_{ret}$ to vary sinusoidally in the $x$ direction by ±0.15 % with a wavelength of 2 km. This approximates the effect of a a change of approximately 0.3 AOD every 1 km. This relatively extreme AOD gradient changes $\zeta_2$ by <4 % in all cases except DRY (not shown). In the DRY LES

run our aerosol gradients induce a factor of 2 change in timestep DRY_7200s and 10 % in other timesteps. This larger relative error may be related to their small spatial variability of TCWV and low values of $\zeta_2$.

This test represents a very large horizontal change in AOD, and since ISOFIT simultaneously retrieves AOD such cases could be identified. Overall, we conclude that for scenes where AOD is of order 0.35 or less, and horizontal gradients are smaller than 0.3 AOD km$^{-1}$, our results will generally be robust to typical aerosol variability. In practice, we would recommend testing each case using coincidentally retrieved AOD fields along with the estimated sensitivity of TCWV$_{ret}$ to AOD to calculate a likely effect of special AOD variability. This may identify cases where aerosol variability affects derived $\zeta_2$.

### 3.3 Solar zenith angle, calculation direction and $\zeta_2$

In our final tests we compare $\zeta_2$ calculated on the TCWV$_{ret}$ fields with SZA changed from 15—60° as a function of the "true" value obtained from the TCWV field. The TCWV$_{ret}$ values are those using each simulation's emulator parameters and include the random error correction, while the true values refer to the columns directly over each footprint with no retrieval error. All use $\Delta x$=40—50 m with fits calculated over $r$=500—1000 m, and calculations are either along the $y$ axis and parallel to the solar azimuth, or along the $x$ axis and perpendicular to the solar azimuth.

Figure 6 shows that there is substantial spread introduced for realistic SZA, with a strong tendency for bias in $\zeta_2$ to increase with SZA when calculated parallel to the solar azimuth. This spread and apparent bias is greatly reduced when the calculation is performed perpendicular to the solar azimuth. In the parallel case there are still significant differences ($p<0.05$) between true and retrieved $\zeta_2$. The $p$=0.05 value threshold is calculated as 1.96 times the standard error of the difference in best-fit parameters derived from the ln($S_2$) as a function of ln($r$) gradient.

Figure 7 shows that both bias and error range expand with SZA when calculating in the solar azimuth direction, but the median (and mean, not shown) bias is eliminated by calculating $S_2$ in the direction perpendicular to the solar azimuth. The spread is also wider in the parallel case, the 5—95 % range of differences relative to the truth is -0.04—0.49 versus -0.20—0.22 when calculating perpendicular, while the standard deviation is 0.18 for parallel versus 0.13 for perpendicular.

These results show that realistic SZA result in path-integrated water vapour structures that have somewhat different characteristics at a 0.5—1 km scale than that of the PCWV$_{PBL}$ over each footprint. This results in additional error to estimated $\zeta_2$, but appropriate calculation strategies that account for solar azimuth angle can reduce the magnitude of this error and suppress systematic error.

### 4 Discussion and conclusions

We have shown that for the $q$ fields simulated in five LES runs of shallow convective PBLs, a novel strategy accounting for solar azimuth eliminates the SZA-induced bias in calculated $\zeta_2$ over $r$ of 0.5—1 km from high-spatial-resolution VSWIR retrievals. This substantially increases the range of applicable scattering geometries for which VSWIR retrievals can be used to estimate spatial scaling statistics. For example, in the airborne case studies of Thompson et al. (2021), only flight lines with SZA<15° could be used, while our method promises unbiased estimates of sub-km $\zeta_2$ for SZA up to 60°. Removal of the bias

is particularly important for applications, since $q$-scaling analyses using spaceborne data typically group or average over many sets of measurements (e.g. Kahn and Teixeira (2009)), and this averaging will not reduce bias in the way that it reduces RMSE. For the first time it also allows calculation of high-resolution $\zeta_2$ statistics over midlatitude and polar areas of the globe that do not experience low solar zenith angles.

This approach should be applicable to any instrument that obtains TCWV from VSWIR with horizontal resolution approaching 50 m, not just EMIT. Operationally, this requires sufficiently long, continuous sampling perpendicular to the solar azimuth. We have not analysed length requirements here, but note that for airborne campaigns this is easily addressed on a flight-to-flight basis. For spaceborne instruments, the sampling will depend on the swath size and orbital geometry. EMIT's approximately 75 km swath (Bradley et al., 2020) spans distances larger than those considered here, and so we expect sampling

to be sufficient regardless of orbital configuration. Similarly, ESA's CHIME contractor notes a 128 km swath, and similar capacities are expected for the missions that address NASA's Surface Biology and Geology (SBG) and Aerosols, Clouds, Convection and Precipitation (ACCP) designated observables. However, we note that it will be necessary to understand the implications of non-uniform footprint size and footprint-dependent errors determined from the on-orbit instrument performance in order to increase the confidence in estimates of $\zeta_2$ from these spaceborne sensors.

Our analysis is based on the retrieval outputs of Richardson et al. (2021), which showed how random retrieval error $\sigma_\varepsilon$ can be identified and removed from $TCWV_{ret}$ fields by exploiting the properties of TCWV $S_2$ at small scales. It also showed that errors in the $T$ and $q$ profiles assumed in the retrieval could affect retrieval sensitivity $a_1=dTCWV_{ret}/dTCWV_{eff}$. We have shown here that to obtain $\zeta_2$ it is critical to remove the effect of $\sigma_\varepsilon$, but that while errors in $a_1$ are the largest error source for estimates of spatial standard deviation, they do not greatly affect the derived $\zeta_2$.

Our results were only determined for areas where the surface type is composed either of mixed vegetation or mixed urban-mineral surfaces, so either a simultaneous surface classification (as provided by ISOFIT) or ancillary surface information would be required. We also identified that horizontal gradients of 0.3 AOD km$^{-1}$ or less only a small effect on derived $\zeta_2$ in most cases, and since ISOFIT simultaneously retrieves AOD it seems likely that cases where this is not true could be identified. However, we highlight further investigation of the effect of realistic AOD structure as a useful future investigation. A major

limitation of this study is that 1-D radiative transfer was used to derive the relationship between TCWV and $TCWV_{ret}$, while clouds can affect nearby clear-sky scenes through 3-D radiative effects. Future work could address this by using a 3-D radiative transfer code (e.g. Evans (1998), Emde et al. (2011)) as the forward model in an improved OSSE.

Ideally this sampling strategy could be field tested, and a strategy to do so would be to perform flights both parallel and perpendicular to the solar azimuth with collocated retrievals of TCWV from VSWIR and an independent instrument that is

not affected by SZA, such as a differential absorption lidar (DIAL) or passive sounding instrument. The non-VSWIR instrument would allow calculation of $\zeta_2$ that is not affected by sunlight path, and the conclusions of this study would be supported if the VSWIR and non-VSWIR estimates showed good agreement for the perpendicular but not parallel flight paths. We highlight the High Altitude Lidar Observatory (Bedka et al., 2021) as a candidate sensor for such an experiment.

For science applications, it would also be necessary to identify which scenes are likely to satisfy our requirements, for example conditions proximate to deep convection may have more substantial above-PBL variability at <1 km scales and result in weaker correspondence between TCWV and $PCWV_{PBL}$. In addition, it may be necessary to identify PBL height, which would require independent information from weather forecast models or reanalysis, *in situ* measurements or other instruments. Furthermore,

users would also need to consider the effect of sampling biases that may depend on cloud fraction or on the presence of non-isotropic variability in features such as horizontal convective rolls (e.g. as explored in Carbajal Henken et al. (2015)). If locations commonly experience meteorological features with a preferential orientation, for example due to orography or a coastline, then this method may not adequately capture its full structure.

Finally, the interpretation of these exponents has not been considered in detail here but future work could proceed either

observationally or theoretically. For an observational study, the relationship between retrieved exponents and other properties, such as later convective initiation could be investigated. A theoretical study would need to apply current understanding of turbulent physics to address precisely the problem of the horizontal variability of vertically-averaged profiles. Further in the future, perhaps multi-angle imaging spectroscopy could provide profiling from VSWIR measurements via computed tomography, which has been demonstrated to retrieve aerosol profiles in some conditions using the Multi-Angle Imaging

SpectroRadiometer (MISR) on Terra (Garay et al., 2016) and upper-tropospheric water vapour profiles from airborne measurements with the Gimballed Limb Observer for Radiance Imaging of the Atmosphere (GLORIA, Ungermann et al. (2015)). We are not aware of any likely short-term space missions that would allow water vapour tomography from VSWIR retrievals at EMIT-like horizontal resolution, with the upcoming Multiangle Imager for Aerosols (MAIA, Diner et al. (2018)) having horizontal resolution similar to that of MODIS. If such an instrument were launched, high-spatial-resolution

tomography would also be subject to the smearing effect from the non-vertical solar path and so may benefit from our proposal, although a detailed study of the particular instrument and retrieval setup would be required.

Our main conclusion is that a novel sampling strategy can allow a breakthrough in space-based measurement of PBL water vapour scaling, and that while individual uncertainties and geographical sampling may vary with the mission, this principle will apply to the array of upcoming VSWIR instruments whose spectra will allow column water vapour retrievals at resolutions

from 30—80 m or better. The resulting statistics offer a new check on the validity of high-resolution atmospheric models and inform sub-grid parameterisations of coarser ESMs.

**Code/data availability:** The MODTRAN radiative transfer code is available at http://modtran.spectral.com/ and requires a licence. The LES profiles used in the emulator development, and the 2-D LES output fields used in this analysis are available at http://dx.doi.org/10.5281/zenodo.5717263.


**Author Contributions:** MTR processed the data and generated the figures. DRT developed the retrieval code (Isofit) used to generate the retrieval emulators. MJK generated and provided the large eddy simulation output. MDL initialised the project and worked with MTR on analysis design and data interpretation. All authors contributed to the conceptualisation, writing and editing.

**Acknowledgments:** This research was carried out at the Jet Propulsion Laboratory, California Institute of Technology, Pasadena, CA, USA, under contract with the National Aeronautics and Space Administration. MR thanks Amin Nehrir and Brian Carroll of NASA Langley for helpful discussion regarding HALO lidar data.

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

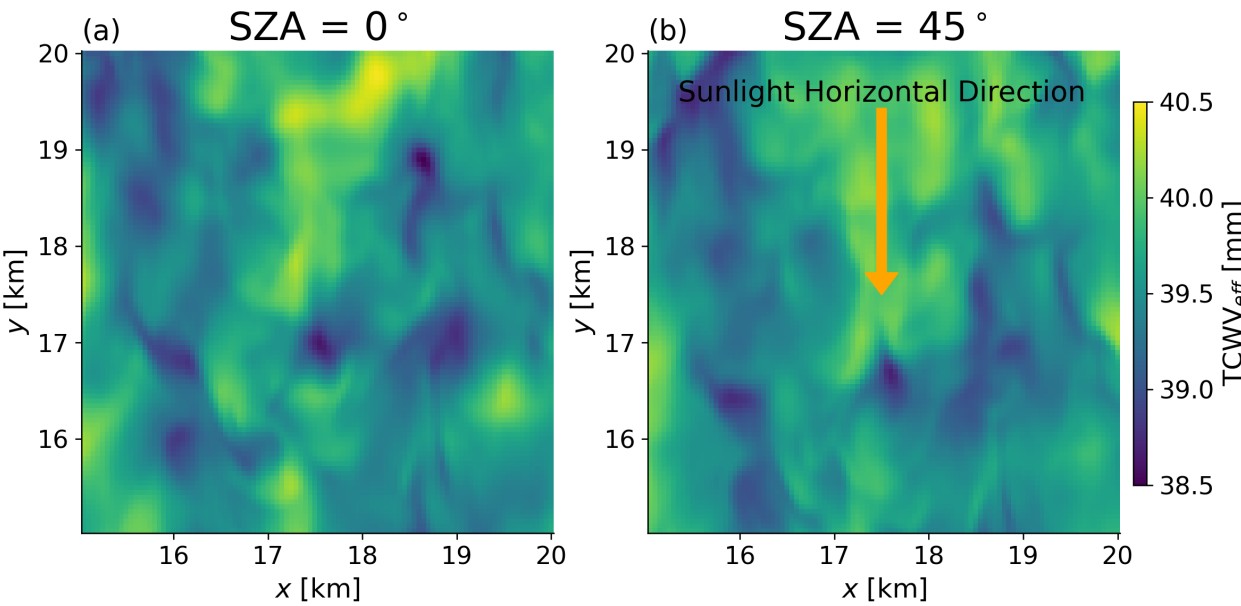

**Figure 1. Integrated water path in a subsection of the ARM_18000s snapshot (a) in vertical columns directly over each footprint (i.e. true TCWV) and (b) encountered by sunlight with SZA=45° viewed from nadir. The yellow arrow indicates the horizontal direction of the downward solar path.**

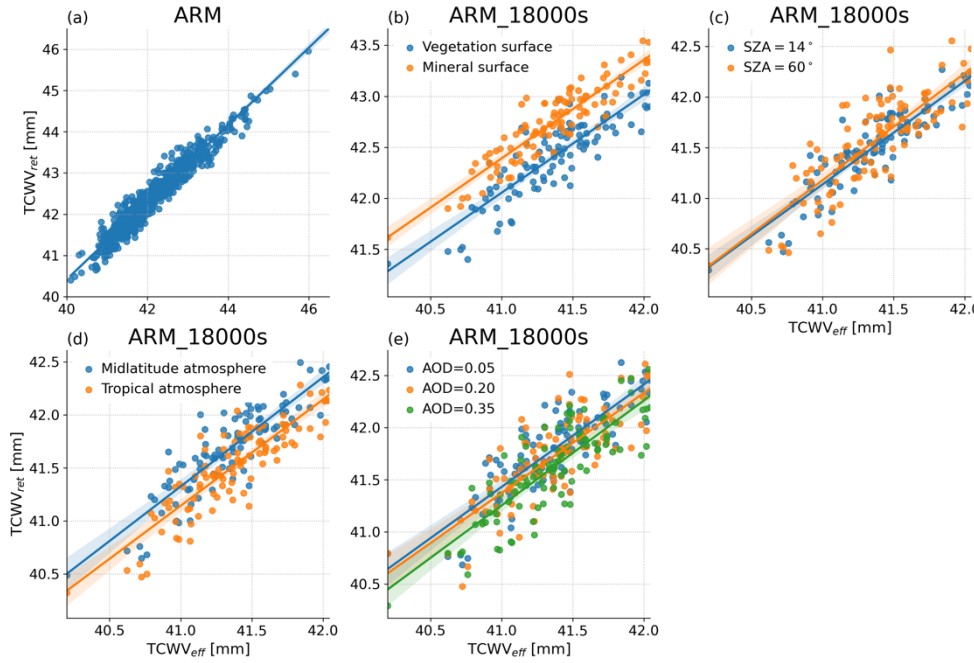

**Figure 2. TCWV$_{ret}$ as a function of TCWV along with emulator fits: (a) for subsets from all ARM snapshots with SZA=45° over a cropland surface. (b) over two typical different surfaces, ARM_18000s only, (c) with different SZA, (d) when changing retrieval's assumed $q(z)$ and $T(z)$, (e) when changing AOD.**

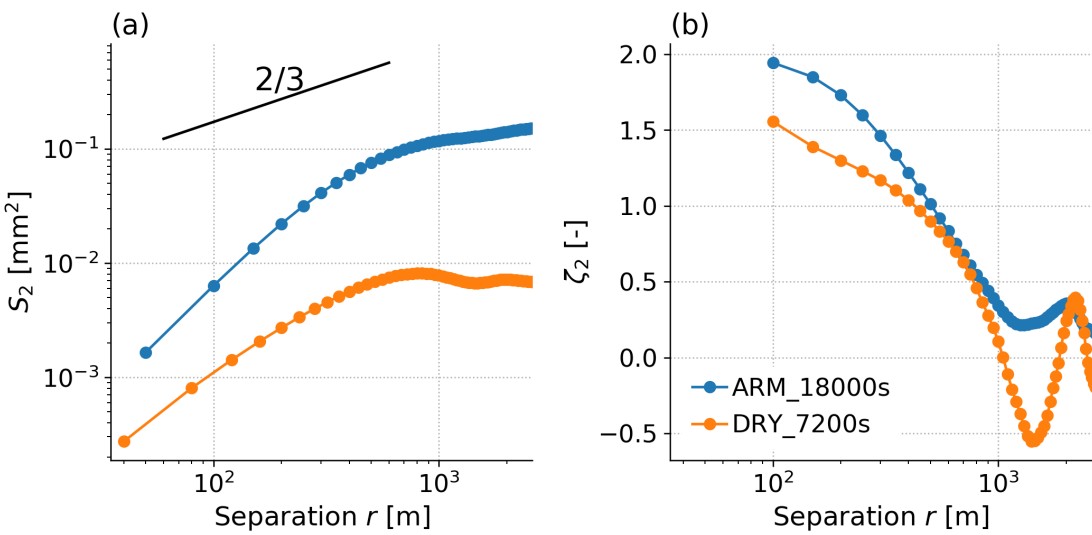

**Figure 3. (a) $S_2$ calculated from TCWV for two LES snapshots as a function of separation distance, with the 2/3 gradient associated with a passive tracer in turbulence following Kolmogorov theory also shown, (b) exponent calculated locally at each separation distance**

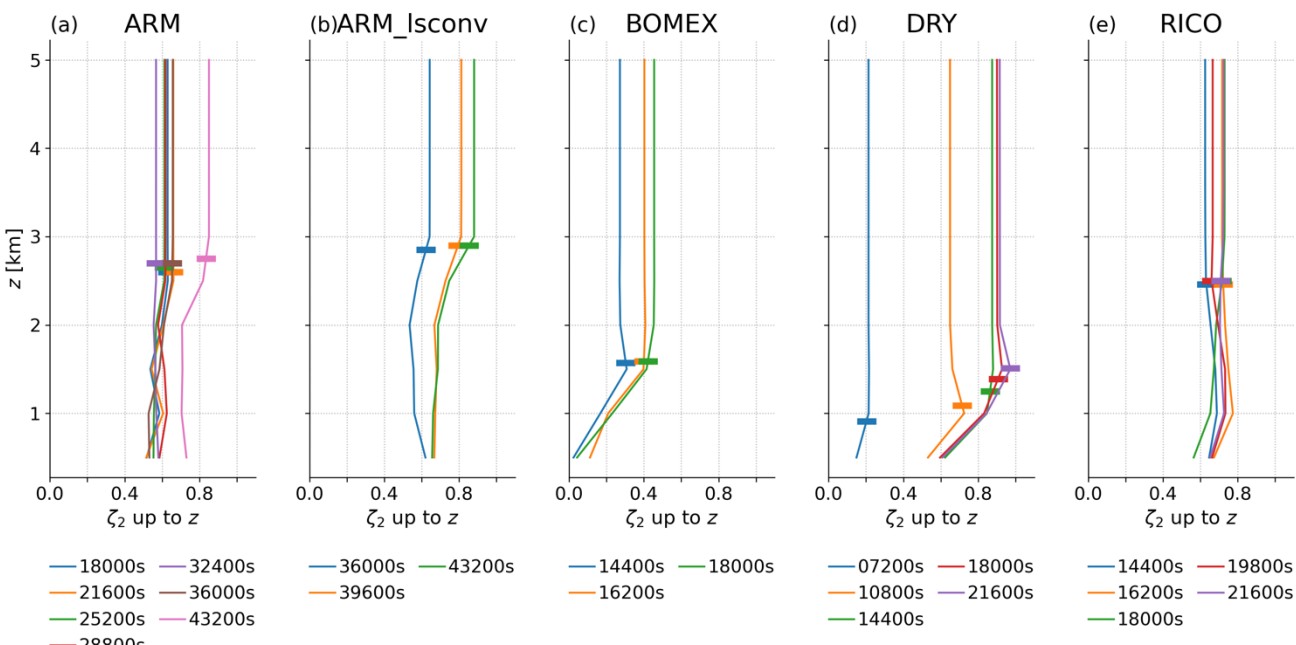

**Figure 4. Calculated structure function exponents using separation distance 0.5—1 km directly on LES output, calculated for integrated water vapour up to each labelled capping altitude, every 0.5 km. The PBL top derived from the location of the maximum vertical gradient in potential temperature is shown as a horizontal bar for each profile.**

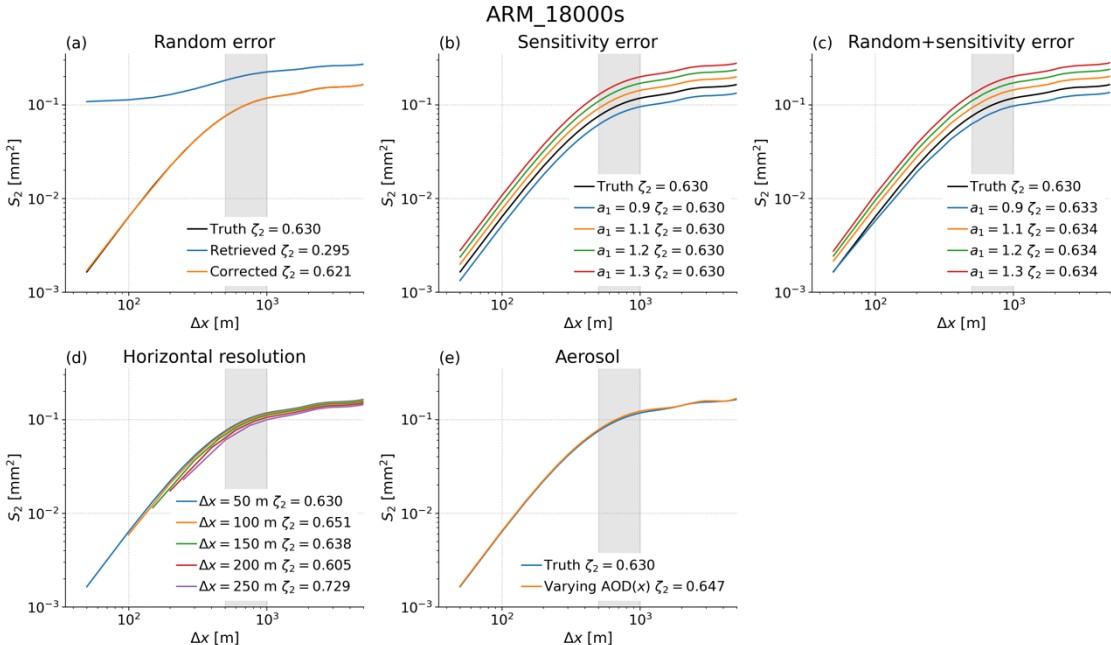

**Figure 5. Sensitivity of derived $S_2$ to retrieval errors and spatial resolution, the legends in each case report the calculated exponent from a fit over separations 0.5—1 km, the region which is shaded grey in each panel. In each case "truth" refers to the value calculated from TCWV at native LES resolution of 50 m. (a) blue shows $S_2$ derived from retrieved TCWV including random error only, and orange the $S_2$ after subtracting the estimated retrieval variance. (b) $S_2$ calculated with no random error, but non-unity sensitivity $dTCWV_{ret}/dTCWV_{eff}$ as defined by the $a_1$ trend parameter in Eq. (8). (c) The result when combining random error and sensitivity error, after subtraction of the random error estimated from each retrieved field. (d) $S_2$ calculated after smoothing the field resolution sequentially to 250 m × 250 m. (e) $S_2$ calculated for a horizontally-varying aerosol field where $TCWV_{ret}$ changes sinusoidally by 0.3 % every 1 km, representing a change of 0.3 in AOD based on the $TCWV_{ret}$ response from Figure 1(a).**

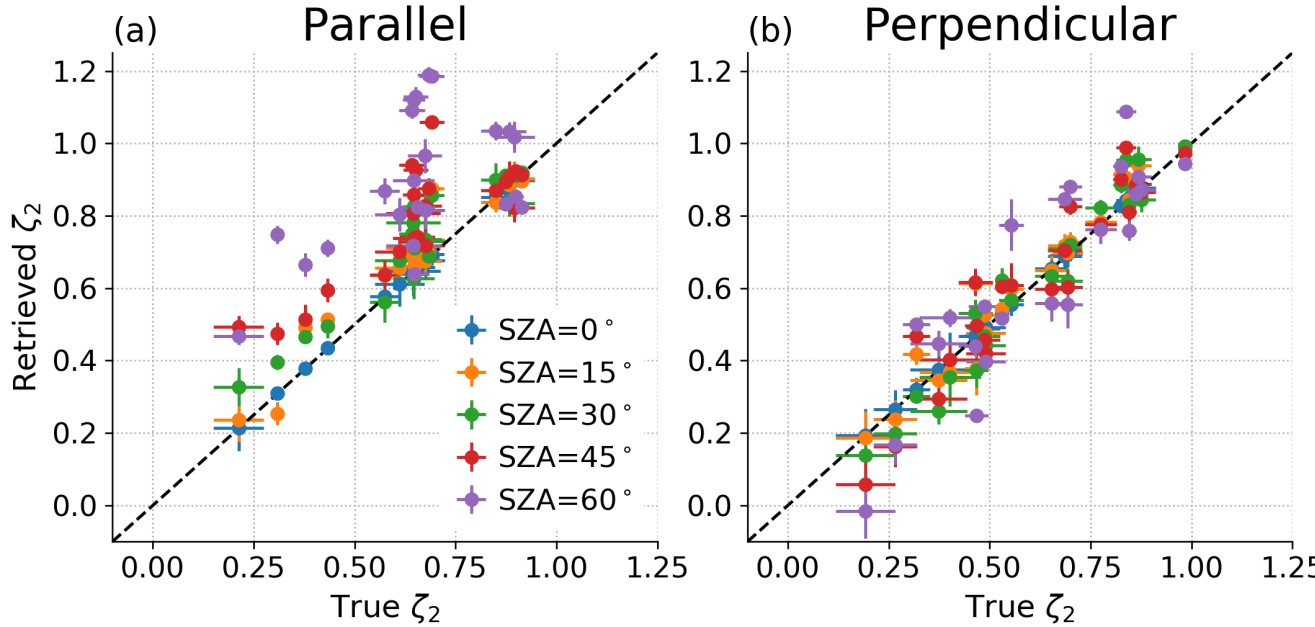

**Figure 6. Estimated clear-sky $\zeta_2$ over separations 0.5—1 km in all 23 snapshots as a function of the true value. The error bars are ±2σ from the trend fit.**

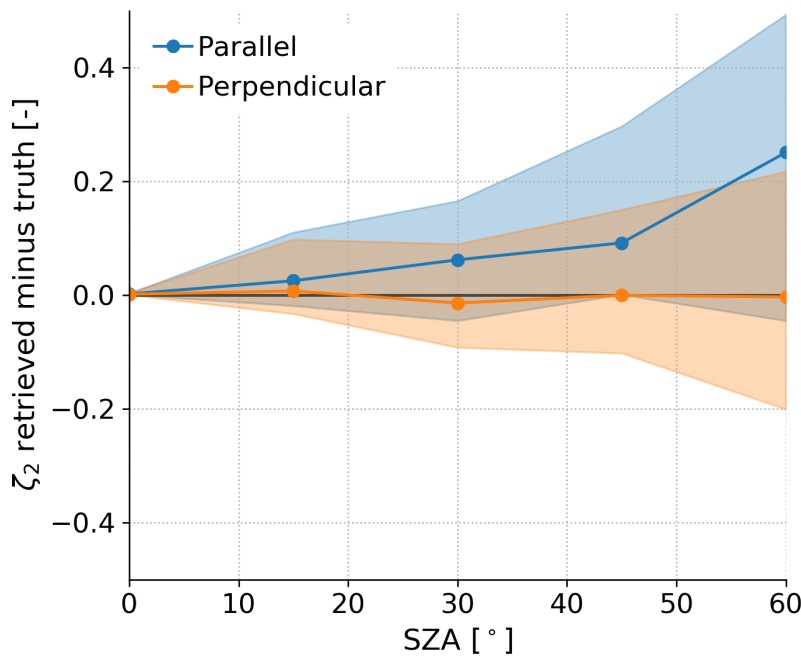

**Figure 7. Median and 5—95 % range of retrieved minus true $\zeta_2$ as a function of solar zenith angle. "Parallel" refers to calculation along the solar azimuth direction, and "Perpendicular" refers to calculation perpendicular to it.**

