# Peer review of "New sampling strategy mitigates a solar-geometry-induced bias in sub-km vapour scaling statistics derived from imaging spectroscopy"

_Atmospheric Measurement Techniques, 2021_

## Author Comment (AC1)

Thank you for a rigorous and challenging review. We disagree with rejecting our paper and respond now to allow discussion and reviewer guidance before the AMTD deadline, in the hope that we can persuade you that there have been technical misunderstandings and that our work is robust and valuable. We see how our current structure causes these misunderstandings and will modify the paper in the meantime.

At the beginning of this project we agreed with your concerns regarding the use of an emulator, and we agree that it is "not justified by this study". To justify it, we performed an OSSE as you describe it, using radiative transfer forward modelling and an optimalestimation inverse retrieval with realistic instrument uncertainty. These results are described in the published companion study Richardson et al. (2021, https://doi.org/10.5194/amt-14-5555-2021). That paper was very, very dense (26 figures inc. supplementary) and we aimed to build on this work with a briefer summary of a result we found very promising for a practical new atmospheric measurement technique.

Your review has made us realise we erred too far on the side of brevity, and simply referencing the prior study did not provide the necessary context. A more explicit title would be something like "Boundary layer water vapour statistics from high-spatial-resolution spaceborne imaging spectroscopy Part II: a new sampling strategy to remove biases in subkm vapour scaling statistics introduced due to the horizontal component of the sunlight path through a horizontally varying water vapour field." But that's ridiculously long! We are open to a better title suggestion but for now intend to add paper content that better describes the title's meaning. We will explicitly link to the OSSE results on which our emulator is based (perhaps with an example figure), better explain terminology and illustrate the solar-smearing bias.

**Methodological details**

We went back and forth about terminology. All VSWIR retrievals obtain path-integrated water vapour *IWV*, but literature phrasing varies and our target physical variable is *TCWV*, referring specifically to the column above a surface footprint. Despite this, much of the literature uses *TCWV* to refer to *IWV* retrievals from VSWIR instruments, e.g. for MERIS1, OLCI2, MODIS3 or TROPOMI4.

Ultimately we selected TCWV with  $TCWV_{ret}$  representing the retrieved value and differences from TCWV include those due to typical errors (e.g. retrieval error) and representation differences (e.g. the fact that it is actually path integrated water).

This is very important and we were not clear in the submission, we will address that with specific text.

Next, we must justify the linear emulator. We are also changing the text to do so, but provide an extended discussion here to hopefully allay the reviewer's legitimate concerns. It is worth

<sup>1 https://doi.org/10.5194/amt-5-631-2012 - "1D-Var retrieval of daytime total columnar water vapour from MERIS measurements"

<sup>2 https://doi.org/10.3390/rs13050932 - "Retrieval of Daytime Total Column Water Vapour from OLCI Measurements over Land Surfaces"

<sup>3 https://amt.copernicus.org/articles/8/823/2015/ - Retrieval of daytime total columnar water vapour from MODIS measurements over land surfaces

<sup>4 https://doi.org/10.5194/amt-13-2751-2020 - Total column water vapour retrieval from S-5P/TROPOMI in the visible blue spectral range

noting that our key result – the use of solar-relative geometry to calculate this spatial statistic of TCWV – is not dependent on using ISOFIT, in principle many retrievals may behave similarly. Minor differences in retrieval performance or emulator fidelity should not alter the fundamental idea that structure function exponents are best calculated perpendicular to the slanted sun-ground path.

That said, we will modify the text to include discussion of previous work justifying the emulator. Firstly, the Richardson et al. (2021) OSSE, on which our emulator is based, used the MODTRAN6 (forward), and ISOFIT (inverse) models so followed the reviewer's standard OSSE description.

Forward and inverse simulations were done for a range of surfaces, aerosol optical depths, and solar zenith angles across each of the LES snapshots.  $TCWV_{ret}$  was well-predicted by a linear fit to the forward-model TCWV.

Several considerations drove us to an emulator rather than a full retrieval. Our radiative transfer has to adequately represent the spectral detail of water vapour absorption across >200 channels from  $\lambda$ =380—2500 nm. We also need sufficient atmospheric layers to capture the vertical structure of absorption line broadening. MODTRAN6 can meet these requirements, but for the 11.8 billion combinations of footprint-SZA-surface-aerosol we originally targeted, serial processing time is many millennia. Our requirement for detailed absorption band spectroscopy makes the computations more expensive than some others that use less detailed spectral/absorption information. Even after turning off outputs including the atmospheric correction data, MODTRAN writes out enough that parallel processing hit a write bottleneck, keeping *parallel* processing time in the millennium range with our resources. Fortunately, we have demonstrated that our optimal estimation retrieval's TCWV performance can be captured very well by a linear fit, which was developed from a subset of footprint combinations and is what we use in this paper. Since the scientific results only care about the level of accuracy of the methods, and our emulator does a good job of representing all of the issues that are relevant for this structure-function analysis, we choose to use it.

Our argument begins with saying that the along-path water vapour is  $q(z_{path})$ , which is distinguished from the real vertical profile q(z).

MODTRAN is plane-parallel so assumes the same atmosphere on the downward and upward paths, albeit the light passes through each at different angles. We found that *regardless of the LES water vapour profile* q(z), and regardless of the  $q(z_{path})$  that implied for the plane parallel simulations, the relationship between the true TCWV and TCWVret was captured by a linear emulator. If this is true for the arbitrary set of profiles we included where differences between  $q(z_{path})$  and q(z) are simply related to SZA, it follow that  $q(z_{path})-q(z)$ differences due to horizontal variability shouldn't destroy the existence of a linear relationship between TCWV and TCWVret either. We anticipate that they will affect the *parameters* that describe that relationship, but our structure-function results are immune to changes in  $a_2$  (intercept doesn't change variance) and  $\epsilon$  (we reliably identify and subtract it, Figure 3a). The slope parameter  $a_1$  is important, and it is the largest error source in estimating spatial standard deviation (a key result from Richardson et al., 2021), but our structure function parameters are actually quite insensitive to realistic values of  $a_1$  (Figure 3b). Richardson et al. (2021) results to support the linear emulator now follow. The first question is: how are realistic profiles of vapour with "TCWV truth" related to retrieved TCWVret? Linearly:

Each of the colours represent different LES cases, and they may have gradients  $a_1 \neq 1$  and which may differ from each other. We hypothesise (with some evidence from retrieval tests) that this is partly due to differences between the mean profiles in each LES, which results in different changes in line broadening as TCWV changes. For this reason we use separate emulators with different parameters for each case.

The linear fit can be characterised with stable parameters using just  $\geq 50$  footprints, provided we sample footprints spanning the LES TCWV range. Here are linear fits with  $\pm 2\sigma$  confidence intervals using different numbers of footprints. In our current paper we used 303—707 footprints per case. Way beyond the number required to obtain a stable fit.

Next, we changed SZA input to the plane-parallel RT and retrieval. For  $\leq$ 45° things are very similar, small (non-significant) changes in parameters occur for SZA=60°, these can be captured by changing the linear parameters, and our structure function results are only weakly

sensitive to these changes. SZA=60° may have a slight change in  $a_1$  (again, it's not significant in this case), but this actually has very little effect on derived  $\zeta_2$  (Figure 3b):

We also checked what happened when we assumed a different vertical profile in the retrieval with the same integrated water vapour. This changes parameters but not the linear relationship: And it is for this reason (one example shown below) that we argue that the  $a_1$  parameter is at least partially related to errors in the shape of  $q(z_{path})$ :

---

## Author Response (AR1)

**Comments to editor**

Thankyou for considering our paper for AMT. We added a lot of content to the paper and much of it answers multiple reviewer comments, so it seemed impractical to quote everything we did in this document. We've therefore just restructured the public review responses: discussion is in red text and description of changes is in magenta.

Our main changes are to include the details requested by reviewer 1 along with additional analysis which we believe to be a reasonable, but strict, test of how AOD spatial variability may affect our results. We show that the effect of AOD is generally small, and point out that in practice we can likely identify cases in which it is not small.

Reviewer 2 recommended rejection in the manuscript's original form, we have now addressed their concerns. This involved adding approximately 2.5 pages of detail, two figures and several equations. In summary: we had actually largely followed the methodology they suggested as the standard way of doing things, as part of our published companion paper. However, we had not explained this (or some other details) well, and so we have now clarified everything they mention.

One concern of theirs that we didn't address was that we did not perform full 3-D radiative transfer for all of the LES outputs. Our approach is nevertheless standard and we argue that it is clearly publishable – we do not have the capacity to do full 3-D simulations and are not aware of any cases where those have been done for the data volumes we require, specifically with very high spectral resolution. We do not believe our paper should be rejected for its use of plane-parallel radiative transfer, which is a common tool in many recently published papers.

Reviewer 1 response in on p2—6, reviewer 2 response on p7—11.

**Reviewer 1**

This well written manuscript deals with future spaceborne imaging spectrometers expected to measure water vapour columns with horizontal resolutions of < 100 m. The authors simulate biases in water vapor scaling statistics that will occur at high solar zenith angles due to a solar light path traversing neighboring pixels. To reduce the biases, the authors propose a sampling strategy perpendicular to the solar azimuth angle. This is evident, and the described bias reduction is what one would expect. The merit of this study, which fits very well to AMT, is a quantification of the expected biases in water vapor scaling statistics. The study still lacks details on assumed measurement uncertainties, see specific comments.

Thanks for taking the time to read and think about our paper. We have added the requested details on uncertainty and additional tests on the effect of AOD.

The main text includes new Figure 2(e) and Figure 5(e) panels showing the AOD results, plus text discussing those, and in the discussion & conclusions. We believe these additions are demonstrate our AOD-relevant conclusions without unnecessarily lengthening the paper. We include extra details in this review response to help the reviewer(s) judge our methodology.

A lot of text and 2 figures were added to respond to reviewer 2 – we kept your comments in mind and responded to them where possible in this added content.

Specific Comments:

1. Spatially nonuniform aerosol distributions (as stated in the abstract) are in my opinion not enough addressed. They probably pose the highest challenges to spectroscopy. On the other hand, they may be difficult to assess, and the resulting biases difficult to quantify. It would nevertheless be of high merit to include them in your model framework and to show some related simulation results in section 2.

**DISCUSSION**

Tucked away in Supplementary Figure 7 of our last paper we showed that $TCWV_{ret}$ isn't that sensitive to AOD, and our emulators were developed with randomised AOD. We anticipated that with typically small(ish) horizontal gradients in AOD over <1 km there would be minor effects on our results from AOD.

However, this paper really should demonstrate this rather than state it, so we added AOD results in the new Figure 2 and added an AOD-results panel to Figure 5.

Our approach was to generate a "very very bad case" with large AOD gradients of order 0.3 $km^{-1}$, and show that it has a small effect on estimated $\zeta_2$. The paper shows the ARM_18000s example, it represents the 22 out of 23 cases where the effect on exponents is small. We expand on this here in case the reviewers are interested. If you only care about further summary of main-text changes, see bolded paragraph at the end.

**Additional detail for reviewers**

The original emulators were fit to forward & inverse simulations with randomised true AOD. We re-ran ARM_18000s and DRY_7200s forward and inverse simulations with profiles where AOD is fixed at 0.05, 0.20 or 0.35 and generated new emulators. We used these emulators *only* for the specific AOD sensitivity tests.

ARM_18000s saw a shifting mean bias of ~0.3 % in TCWV$_{ret}$ when AOD changes from 0.05—0.35. The changes are proportionally smaller for DRY_7200s but include changes in gradient. For our first test we modified our emulators to make the gradient and intercept into functions of AOD and fit them:

$$TCWV_{ret} = a_1(AOD)TCWV_{eff} + a_2(AOD) + \epsilon$$

[Figure]

The panels above show the fits that provide sets of $a_1$ and $a_2$ values given AOD in [0.05, 0.20, 0.35 ]. For any AOD we simply linearly interpolate between those to provide our AOD-dependent emulator. The next issue was to decide what sort of spatial variability in AOD to test.

We decided on a rather extreme case: changes from 0.05—0.35 every 1 km, represented by a horizontally-varying sinusoid with period 2 km in either the *x* or *y* direction. Below is true TCVW, emulated TCWV and the difference between them for DRY_7200s and ARM_18000s at SZA=0°. The horizontal waviness from the spatial AOD structure is obvious in (c,f). We did not add random error ($\epsilon$) here, but the paper shows that we can identify and remove its effect on $S_2$. (ARM masked values are those that are cloudy or shaded at any of our selected SZAs).

[Figure]

Next, we address how $S_2$ and $\zeta_2$ respond, firstly in just these two snapshots. The figures below show they are offset: this is primarily because of $a_1$ – it scales $S_2$ as shown and discussed in main paper Figure 5(c). Note that panel (a) below differs from the new Figure 5(e) for reasons discussed below.

This DRY case is the outlier value with the lowest $\zeta_2$ of all snapshots, and it's clearly due to the "dip" near ~1 km separation suppressing the gradient. As discussed in the paper, we do not investigate the detailed dynamics of the LES cases here but are only interested in how well we can retrieve the property. In the ARM case the change in $\zeta_2$ is negligible (0.63 vs 0.63), but in DRY_7200s changes greatly from ~0.2 to ~0.4.

[Figure]

The DRY_7200s difference occurs because the true S2 structure shows a decrease in variance near ~1 km while our AOD-induced variability has a maximum effect at 1 km separation. We wanted to test all snapshots but didn't have time to process the forward and inverse simulations needed to generate individual emulators, so we used another approximation. In percentage terms, the ARM_18000s case shows the largest effect: a 0.1 % change in $TCWV_{ret}$ per 0.1 change in AOD, so we used that as the basis of our next test.

We added a simple treatment of aerosol to all emulated snapshots: a sinusoidal variation in $x$ or $y$ of $TCWV_{ret}$ with an amplitude of ±0.15 % of the field mean and a wavelength of 2 km. This is just like the test above, except there is no change in $a_1$. For Figure 5(e) the test uses $a_1 = 1$, so the $S_2$ lines lie atop each other rather than have the offset seen in the above figure.

This results in a change of 0.3 % in $TCWV_{ret}$ every 1 km to represent a change of ~0.3 in AOD over that distance. The next figure shows how the difference is negligible when calculated perpendicular to the aerosol variation (see panel a) but when calculated parallel to the aerosol variation (panel b) there are changes. The direction dependence is rather like our sensitivity to solar azimuth. From this figure the DRY_7200s case is the outlier of the set: changes in non-DRY snapshots are always <4 % in magnitude, for the DRY snapshots except DRY_7200s it is ~10 %.

One nice thing about ISOFIT is that we would have $TCWV_{ret}$ and retrieved AOD fields, so we could "back calculate" the likely effects of AOD on retrieved $\zeta_2$ and flag cases where it might be important. We have referred to this briefly in the new text.

[Figure]

**CHANGES TO MANUSCRIPT**

In summary: we hope this review response persuades the reviewer that we did our due diligence for AOD. In the main paper we have added the following:

i. Figure panel 2e showing how $TCWV_{ret}(TCWV)$ changes as a function of AOD in ARM_18000s
ii. Figure panel 5e showing how $S_2$ changes with a strong spatial variation in AOD in ARM_18000s
iii. Text summarising changes in $\zeta_2$ are generally small even with relatively large AOD gradients, and noting that while it has a large effect in one snapshot, we could use the ISOFIT retrieved fields to flag cases where this is likely in practice. AOD-relevant text is in Sections 2.2, 3.2 and 4.

We think these changes strike the balance between being sufficient and brief.

2. In section 2.2 you define parameters related to assumed measurement uncertainties and biases. Since they are used throughout the study, it would be good to describe them better here, perhaps including a figure which illustrates sensitivity (a1) and bias (a2). In addition, you should be more specific concerning the impact of aerosol layers (comment 1), and concerning probable error correlations between (for example) surface albedo and aerosol concentration variations. Finally, can you assess the impact of simulation idealizations and simplifications which you have likely undertaken?

**CHANGES TO MANUSCRIPT**

Our completely restructured Section 2 aims to address these comments. The new Figure 2 shows how changes in individual properties (SZA, surface, AOD) change the response. Our new text emphasises that we propose retrieving only over mixed-vegetation or mixed urban-mineral surfaces, which have similar characteristics. We would also have near-constant SZA in our samples. Since surface type and SZA effects will be near constant if our method is followed, we do not think it is important to include covariance with AOD.

Covariance between the LES-simulated $q(z)/T(z)$ and aerosol is implicitly included in our development of the emulators, so we believe we have now displayed the things that matter for our application.

Technical Comments:

p.5 line 3: "with CWP calculated in the same manner as the TCWV": also pressure-weighted? Likely not.

**DISCUSSION**

Actually yes, given the units we had. This text has now been deleted though.

**CHANGES TO MANUSCRIPT**

Section 1 new text explains our different path definitions for water vapour and Section 2.2.2 describes the calculation. In hindsight the "pressure weighting" is redundant information – there is only one way to convert our path-traced values into units of mm so we removed that term.

p.7 line 6: "random errors that we estimate": please give examples (numbers) for these errors, in % of the TCWV.

**CHANGES TO MANUSCRIPT**

Added. We slightly rephrased and added: "(0.6 % of TCWV in this case)". This was the $\sigma_\varepsilon$ discussed in the emulator equation, and the OSSE range was 0.5—0.7 % of mean TCWV depending on the LES case (values in mm are in Richardson et al., 2021, Table 2).

**Citation**: https://doi.org/10.5194/amt-2021-163-RC1

**Reviewer 2**

This manuscript documents an OSSE-type (Observation System Simulation Experiment) study of how the high-spatial resolution spectroscopy observation of total column water vapor from satellite observations should be sampled to understand the horizontal variability and structure of water vapor in the planetary boundary layer (PBL).

Topic of this study is important and suitable for the AMT. However, the manuscript suffers from several major issues and significant flaws as pointed out below. Its methodology (i.e., using a simple emulator instead of full simulator) is not justified and has serious potential problems. No causes and underlying physics are provided for the "solar-smearing bias", which is a key finding of this study. Even though the methodology is problematic, and the results are not explained, the authors still try to propose a universal "new sampling strategy" to the current and future high-resolution spectroscopy sensors. This is overreaching the say the least and could be misleading.

**Based on these considerations, I strongly recommend rejection of this manuscript. It this study were published, the "emulator method", the "solar-smearing bias", and "new sampling strategy" could be cited again and again as if they were correct. But they are not, at least not justified by this study.**

We thank the reviewer for their rigorous approach, which made it obvious that we had not been explicit enough about important details. Some other readers could clearly be confused by our original submission, so we have added two new figures and text to hopefully avoid this confusion. We believe that all reviewer concerns are now addressed.

Below we reference page and line numbers, which refer to those in the *red lined/track changes* version of the manuscript.

Major problems:

- The first major problem of the manuscript is the lack of important details on the methodology and the discussions are often too short and unsatisfying.
  - Although the concept of "total column water vapour" (TCWV) appears to be simple, the retrieval process can be quite complicated and involves many technical details, especially at high-spatial resolution. For example, when water vapor has both strong horizontal variation and vertical gradient, the solar-viewing geometry will become important because the path-integrated water vapor can be significantly different from the TCWV, depending on how instrument geolocation/collocation is done. In such situation, observations from different angles need to be de-convoluted to re-construct the horizontal and vertical structure of water vapor. The manuscript briefly mentioned this issue in section 2.2 and 3.1 but the discussion is far from clear or satisfying. For example, it is mentioned "TCWVret from input TCWV, which is in fact the integrated water path along the solar path". But how is the "path-integrated water path" converted back to the TCMV (only times a cosine factor?)? Is the definition of TCWV dependent on solar and/or viewing angle? Although Figure 2 provides some information on the vertical variation of water vapor of

the cases used in this study, the corresponding discussion in Section 3.1 is so brief (only one sentence) and obscured that it only raises more questions than answers. In particular, it is hard to tell how the author could "confirm that our derived values are indeed representative of bulk PBL statistics" from the figure, when there seems to be significant vertical variation of epsilon in the PBL.

**DISCUSSION**

Here is where we realised we explained some of our most important method details very poorly.

Our "solar smearing" refers to how the non-vertical solar path through the atmosphere, which cuts through a complicated 3-D field, "smears" the apparent 2-D retrieved TCWV maps. This is explained in new content as detailed below.

All current imaging spectrometer retrievals of atmospheric water vapor ignore this effect when reporting the TCWV; in other words, they do not perform the tomographic reconstruction that the reviewer rightly calls for.  To our knowledge, our study is the first that attempts to account for these effects with this class of instruments.

Please see response to next comment for details on how the path-dependence is included in the radiative transfer calculations behind our emulator, and how it is also accounted for in the emulator inputs.

**CHANGES TO MANUSCRIPT**

We have now added detail in Section 1 on p3L30 onwards and Figure 1. We added equations and description that relate path integrated water vapour (PIWV), real TCWV, and the reported "TCWV" retrieved by VSWIR instrument. We use TCWV terminology for consistency with other VSWIR work, citing 6 papers that call their retrievals "TCWV", although we use subscripts to differentiate properties. In particular, we refer to "effective" TCWV:

$$TCWV_{eff} = \frac{PIWV}{\frac{1}{\mu} + \frac{1}{\mu_0}}$$

Which is the TCWV that would provide the same PIWV given the solar/view geometry. The PIWV is determined from tracing the solar ray through the atmosphere, and our retrieved $TCWV_{ret}$ are estimates of this $TCWV_{eff}$.

Figure 1 compares TCWV and PIWV when SZA=45° in a small part of an LES snapshot domain. An arrow indicates the horizontal component of the solar path; it is visibly obvious that the IWV field is like the TCWV field but "smeared" or "smoothed" in the horizontal. This is simply the result of the solar downward path being diagonal, rather than vertical.

[Figure]

We also changed "confirm that our derived values are indeed representative of bulk PBL statistics" to "…are indeed representative of $\zeta_2$ derived from $PCWV_{PBL}$" since we retrieve the value derived from bulk PBL water vapour, not the average of exponents calculated at higher vertical resolution.

  o Some other technical details are also missing. For example, how cloud mask is applied? Is it dependent on the sun-viewing geometry? If cloud mask is independent of sun-viewing geometry then there is apparently an inconsistency between the use of path-integrated TCWV and use of path independent cloud mask. Is the 3-D radiative transfer considered in the simulation or emulation? Previous studies have noted the "halo effects" of cloud in the so-called twilight zone. How are these 3-D effects of cloud treated in the study? Are they simply ignored (i.e., using 1-D RT model), or removed by cloud masking (then how?) or considered in the simulation?

**CHANGES TO MANUSCRIPT**

We have split Section 2.2 into two sections. Subsection 2.2.1 is almost entirely new text which describes the OSSE approach: we used 1D RT (MODTRAN) and an optimal estimation inverse method (ISOFIT) to derive the emulator, for which we found Eq. (4) (now Eq. 8) was an adequate representation.

Subsection 2.2.1 also refers to a new Figure 2, which demonstrates the linear relationship we assert for the emulator, and shows how the parameters may change with surface, SZA, retrieval-assumed $q(z)/T(z)$ and AOD.

[Figure]

Subsection 2.2.2 now expands on the ray tracing we used to generate and explicitly states that we use the TCWV$_{eff}$ derived from ray-traced PIWV through the 3-D LES field as input to our emulator. We hope that, in combination with the additional detail in Subsection 2.2.1, this is now clearer. Given this context we believe readers can now understand how our cloud/shadow mask is generated in a completely analogous way: "The same ray-traced calculation is repeated with cloud water $q_c$ to obtain cloud water path (CWP). Footprints are then flagged as cloudy or shaded when CWP$>1\times10^{-3}$ mm…".

We agree with the reviewer that 3-D radiative effects could be very important. These would be implicitly addressed by our suggested airborne experiment but we were remiss in not specifically mentioning it. As noted in the new 2.2.1 text, we did not have the computational resources for 3-D RT across all of our desired cases, especially given our very high spectral resolution requirements. Section 4 now mentions that 3-D RT forward modelling is a good way to improve this: we reference papers behind SHDOM and MYSTIC here.

- The use of a very simple retrieval emulator is not justified and raises many questions.
  - OSSE type of studies often use a "retrieval simulator" consisting of a "forward" RT simulator and an "inverse" retrieval simulator. The simulator should be as "realistic" as possible in comparison with the real retrieval to faithfully capture the influences of various factors on the retrieval. In contrast, this study only uses a seemingly naïve retrieval "emulator" (i.e., equation 4) and the only reason to justify this is "due to computational constraints". This "emulator" skips both the RT simulation process and the retrieval simulation step, and directly connects the retrieval to the input fields in a very simple way (linear). There is no discussion on the accuracy of this emulator in comparison with the "full OSSE simulator" if there is one. As a result, it is unclear if the artifacts in the "retrieval" is meaningful or simply due to the inadequacy of the emulator. It is also hard to imagine what kind of "computational constraints" the authors are referring to. This is a case study based on a handful of LES scenes. Many previous studies have performed full RT simulations, even 3-D

RT simulations, based on LES scenes. How and why is the RT or retrieval simulation in this study so computationally expensive?

**DISCUSSION**

Please see responses above. The reviewer describes what we believe to be the "correct" way, which is indeed what we did in Richardson et al. (2021). The new 2.2.1 text describes the previous OSSE from which Eq. 8 (originally Eq. 4) is derived, and the new Figure 2 (above) shows some evidence that should persuade readers to provisionally accept the linearity between $TCWV_{eff}$ and $TCWV_{ret}$. More details are in our previous publication.

- The solar-geometry dependent retrieval bias in section 3.3 is interesting. However, I tried hard to find some explanation of the causes and underlying physics but didn't find any. There is neither any reference to previous studies or discussion on whether this phenomenon had been discovered before or completely new. **The authors didn't even bother explaining why this bias is called "solar-smearing" effect.** The word "smear" only occurred twice in the manuscript, one in the title and the other in the conclusion.

**DISCUSSION**

Again our failure to sufficiently link back to Richardson et al. (2021) caused confusion.

**CHANGES TO MANUSCRIPT**

The new text in Section 1 mentioned above describes the physical principle, namely the solar path through a 3-D field and explains why we pick the term:

"The **Error! Reference source not found.**(a) to **Error! Reference source not found.**(b) differences show a smoothing or smearing in the *y* direction, so we refer to these solar-geometry induced changes as the "solar smearing" effect." (figure 1 is the first shown in this review response)

- Event though the "solar-smearing" effect is completely unexplained (and is based on highly questionable methodology), the authors still recommended the "new sampling strategy" to many current and future sensors. This totally unacceptable to me.

**DISCUSSION**

The reviewer was right to be cautious given the lack of clarity in our original submission, but we are convinced that the concerns you rightly raised are now addressed. After all, with the exception of 3-D radiative transfer, we actually performed the calculations in ways that you proposed (an OSSE with forward an inverse models) and directly accounted for issues you raised (the complex 3-D structure of the water vapour field).

**CHANGES TO MANUSCRIPT**

Regarding 3-D radiative transfer, this was a limitation of our available tools and computational resources but it is a very good suggestion which we now mention in the discussion & conclusions.

---

## Referee Report (RR1)

**On the review of the manuscript "New sampling strategy removes imaging spectroscopy solar-smearing bias in sub-km vapour scaling statistics" by Richardson et al.**

I didn't read the whole paper. I read the abstract, both reviews and the responses.

I completely agree with RC2 about possible 3D effects on water vapor retrievals. They are important, indeed, and in some cases the effects could be really strong leading to incorrect retrieved atmospheric water vapor. However, I disagree with rejection of the manuscript. Talking about the transition zone between cloudy and clear air, retrieval of water vapor is a difficult problem for all observation: satellite, ground-based and even airborne. We recently published a paper:

Wen G. and A. Marshak. 2021. Precipitable water vapor variation in the clear-cloud transition zone from the ARM shortwave spectrometer. *IEEE Remote Sens. Lett.*, doi: 10.1109/LGRS.2021.3064334.

We didn't use 3D retrievals though we applied 3D radiative transfer to check it. I know another (an old one) MODIS WV retrieval papers

Gao B.-C. and Y. J. Kaufman, "Water vapor retrievals using moderate resolution imaging spectroradiometer (MODIS) near-infrared channels," *J. Geophys. Res.*, vol. 108, no. D13, p. 4389, 2003, doi: 10.1029/2002JD003023.

Of course, it didn't use it either.

The authors of the paper gave a detailed response. If they clearly state in the paper that the 3D radiative transfer effects could be important, but they didn't account for them here, it will be appropriate (in their reply they wrote that it is planned to account for the 3D effects in their next airborne experiment). I think that both 1D and 3D retrieval papers are needed; the 3D-based papers will show problems with 1D approaches and the ways to correct (or, at least, to mitigate) these effects but it takes time, especially if we want to make them operational.

I absolutely agree that "*the simulator should be as "realistic" as possible in comparison with the real retrieval to faithfully capture the influences of various factors on the retrieval*." But I also believe that a naïve simulator is needed as well. Long time ago, we ran (with Warren Wiscombe and Anthony Davis) 3D radiative transfer with a "naïve" Henyey-Greenstein phase function rather than a realistic one; this helped us to focus on other 3D problems rather than droplet scattering.

To summarize, I agree with most of the reviewer 2 comments, but I disagree with the conclusion. I don't think that the manuscript should be rejected if it discusses the possible problems with the suggested approach.

Alexander Marshak

---

## Author Response (AR2)

We again thank the reviewers for their time analysing our manuscript and response. While we are disappointed that reviewer 2 still raised strong concerns, we are convinced that this is primarily due to misunderstandings and have made further changes to reduce the risk that new readers would share these misunderstandings.

Regarding the physical reasoning for our results, we have added a supplement to provide simpler examples and more explanation but believe that most readers would intuit the concept from the main text alone.

We wholly agree with reviewer 2 that accounting for 3-D radiative effects, particularly scattering into nominally clear-sky footprints from nearby clouds, is important. However, this paper overcomes a fundamental bias caused by the direct solar path alone, and so is an important step. We are gratified that reviewer 3 agrees that publication is still justified given that we describe these limitations and identify how to experimentally address them. For further specificity we re-titled the paper "New sampling strategy mitigates a solar-geometry-induced bias in sub-km vapour scaling statistics derived from imaging spectroscopy", edited the abstract and added some later text. The title is unwieldy, but still shorter than some other recent AMTD submissions.

We have changed one sentence requested by reviewer 1. Given the judgment by reviewer 3 we believe our paper has now fully addressed review concerns.

In addition, line- or marker styles were changed in Figures 5 and 6 after a colour-blindness simulator suggested some groups were hard to tell apart. The LES outputs necessary for reproduction have also been uploaded to Zenodo and linked in the paper. Finally, the copyright statement has not been removed from the main file – I need a form that the JPL copyright office can use to transfer copyright.

Reviewer comments stay in black, our general comments are in red and change descriptions are in magenta.

Reviewer 1
"manuscript version 3, p. 9 line 24:
"...our aerosol gradients induce a factor of 2 change in DRY_7200s and 10 % in other timesteps"
This sentence is not clear.
In which parameter is the change observed: TCWV or zeta_2 ?
Include "LES timesteps"."

We broke up this sentence and rephrased:
"In the DRY LES run our aerosol gradients induce a factor of 2 change in $\zeta_2$ in timestep DRY_7200s and 10 % in other timesteps. This larger relative error may be related to their small spatial variability of TCWV and low values of $\zeta_2$."

This is the only comment from reviewer 1.

**Reviewer 2**
Problems with the "Emulator": Although more details are added about their "Emulator" in the revised manuscript, the fundamental problems with it still remain. The authors argue that they can simply "emulate" the impacts of 3-D radiative transfer effects and the parallax effect

on TCWV retrieval without using the advanced 3-D radiative transfer model. I simply do not see how this is possible. The author can certainly derive a PIWV based on Eq. (4) and a LES filed, but that is not what the instrument observes (which is radiance or reflectance). A plane-parallel RT model (MODTRAN) is used to "generate the true forward radiance spectra" (page 6 line 9) in this study. But isn't the "true radiance spectra" the observed spectra that are affected by the 3-D radiative transfer effects and the parallax effect? How could a 1-D model generate the "true" observation? It is mentioned that "in this case, TCWVeff=TCWV" (page 6 line 12). To have TCWVeff=TCWV, we need to have Eq. (6) = Eq. (7) (BTW, Eq. (7) is wrong), which implies PIWV=PIWV_uniform in these simulations. But isn't the whole idea here is to simulate an ununiform PIWV? In summary, to me there are fundamental problems to the methodology of this study (emulator) which make the results based on it highly skeptical.

From reviewers 1 & 3 we judged that our description of the limitations and how future work could address them was sufficient to support publication regarding 3-D RT. Nevertheless, we made further changes to be more precise in our descriptions.

We suspect we are referring to the effect that the reviewer calls "parallax", but a standard definition of parallax is "a displacement or difference in the apparent position of an object viewed along two different lines of sight" (Wiki) but that isn't what's happening here with our single line of sight.

We show that the direct solar path, on its own, causes an insurmountable bias in derived $\zeta_2$ when calculated in the solar azimuth direction. We have changed text (see below) to be explicit that we address *only* this bias, and that a perpendicular calculation overcomes it.

Our emulator is developed (or trained) using plane-parallel radiative transfer, effectively assuming horizontally-uniform water vapour fields. However, when applying our emulator, the input TCWV$_{eff}$ is derived from tracing through non-horizontally-uniform fields. There aren't substantial differences in the shape of $q(r_{\downarrow\uparrow})$ between the training (plane-parallel) and forward-simulation (3-D field) sets so this emulator should work. We believe that our description is already clear enough on this, based on other review comments.

The reviewer also seems concerned about a 3-D factor we do not address, namely scattering into nominally clear-sky footprints from other parts of the sky, primarily nearby clouds. However, we note this limitation and reviewer 3 concludes that this note is acceptable. Text is changed to emphasise this distinction between direct-beam and out-of-footprint factors:

The abstract has been rephrased, including with new text:
…accounting for realistic non-vertical sunlight paths. We trace direct solar beam paths through large eddy simulations (LES) of shallow convective PBLs, and show that retrieved 2-D water vapour fields are "smeared" in the direction of the solar azimuth. This changes the horizontal spatial scaling of the field primarily in that direction…
And:
…By only considering the direct beam we neglect 3-D radiative effects, such as light scattered into the field of view by nearby clouds. However, our proposed technique is necessary to counteract the direct-path effect of solar geometries and obtain unique information about sub-km PBL $q$ scaling…

And in Section 1 the added text includes:

…We show that, in a set of 23 LES snapshots, the non-vertical direct-beam path prevents accurate retrieval of sub-km $\zeta_2$ when standard methods are naïvely applied…

And:

…Diffuse sunlight is handled through a plane-parallel radiative transfer approximation, which means that complex 3-D radiative effects are neglected. In clear-sky areas near clouds, 3-D effects can brighten observed spectra (Várnai and Marshak, 2009), with induced biases of order ~0.25 % for VSWIR column $CO_2$ retrievals (Massie et al., 2021). The consequences for hyperspectral TCWV retrievals at 30—80 m horizontal resolution are not currently known, although the effect on retrieved $\zeta_2$ would depend on the spatial scaling of these TCWV biases…

• As I mentioned in the first round of review, this paper only presents a phenomenological study of the simulation results, it provides little, if any, explanation of the underlying physics. I think this is due to the use of "emulator" which cannot provide any meaningful explanation of the simulation. Without a solid physical interpretation, these results are highly skeptical to me.

• In section 3.3, the authors found significant differences between the "parallel" and "perpendicular" direction retrievals. Again, there is no explanation of why. Nevertheless, the lack of physical explanation does not prevent the authors from proposing a new sampling scheme to all the possible sensors. To me this result is skeptical to say the least. Any point on the surface can be considered to be either "parallel" or "perpendicular" (or both parallel and perpendicular) to sunlight direction, isn't it? What is the fundamental difference between any two points in terms of geometry?

Our interpretation is that these two bullets boil down to "there is no physical explanation". We believed that we provided this explanation, and reviewer 1 correctly interpreted this as being "due to a solar light path traversing neighboring pixels" and found that the approach was "evident" with a "described bias reduction is what one would expect".

Our explanation was clear for some types of readers, but clearly not others. To try and be as widely understood as possible, we added a supplement and referred to it in the main text: for a simplified illustration of the physical principles behind why our strategy is anticipated to reduce biases in $\zeta_2$, see Supplementary Figures 1—3.

The new supplement and its figures show a simpler idealised situation, including a visualisation of the smearing effect and explicit calculations of spatial statistics: means, second-order structure functions, and $\zeta_2$. We also make an analogy to "motion blur" in image processing, which is pretty widely known and shares much with our problem. For example, the appearance of objects in an image is "smeared" or "blurred" when there is relative movement between the imager and objects within the plane that is normal to the image-object vector.

In particular, objects are apparently "blurred" or "smeared" *preferentially in the direction of motion*, and the apparent spatial structure (which can be captured by statistics), is *also preferentially affected in the direction of the blurring*. Therefore, the direction in which you calculate statistics can matter.

Our problem is analogous: the TCWV field is "smeared" in the solar azimuth direction, in a way that shares features with motion blur. We use the term "smearing", rather than "blurring"

since it is a synonym for the visual effect but "blurring" is associated with motion. As mentioned above, we do not call it "parallax" since we only have one line of sight.

We thought that this was an intuitive result, but our initial description was too terse. We think the added text following the initial review round, combined with the supplement, strike a good balance since for many readers (such as reviewer 1), the process should be intuitive and obvious, and for them the supplementary text would labour the point and distract from the main results.

Overall we now believe that with the supplement we have addressed potential confusion in a balanced way.

**Reviewer 3**
Reviewer 3 largely addressed the issues raised by reviewer 2 and judged that our response was adequate for publication.

They raised some further good points so we added two new citations; Varnai & Marshak (2009) and Massie et al. (2021). They are in the reviewer 2 response text above, and although they do not match the ones recommended by the reviewer, we believe they are the most appropriate references for the point we are making.